# 'Climate Response Functions' for the Arctic Ocean: a proposed coordinated modelling experiment

John Marshall[1], Jeffery Scott[1], and Andrey Proshutinsky[2]

[1]Department of Earth, Atmospheric and Planetary Sciences, Massachusetts Institute of Technology, 77 Massachusetts Avenue, Cambridge, MA 02139-4307 USA

[2]Woods Hole Oceanographic Institution, 266 Woods Hole Road, Woods Hole, MA 02543-1050 USA

*Correspondence to:* John Marshall (jmarsh@mit.edu)

**Abstract.** A coordinated set of Arctic modelling experiments is proposed which explore how the Arctic responds to changes in external forcing. Our goal is to compute and compare 'Climate Response Functions' (CRFs) — the transient response of key observable indicators such as sea-ice extent, freshwater content of the Beaufort Gyre, etc. — to abrupt 'step' changes in forcing fields across a number of Arctic models. Changes in wind, freshwater sources and inflows to the Arctic basin are considered. Convolutions of known or postulated time-series of these forcing fields with their respective CRFs then yields the (linear) response of these observables. This allows the project to inform, and interface directly with, Arctic observations and observers and the climate change community. Here we outline the rationale behind such experiments and illustrate our approach in the context of a coarse-resolution model of the Arctic based on the MITgcm. We conclude summarising the expected benefits of such an activity and encourage other modelling groups to compute CRFs with their own models so that we might begin to document their robustness to model formulation, resolution and parameterization.

## 1   Introduction

Much progress has been made in understanding the role of the ocean in climate change by computing and thinking about 'Climate Response Functions' (CRFs), that is perturbations to the climate induced by step changes in, for example, greenhouse gases, fresh water fluxes, or ozone concentrations (see, e.g. Good et al. 2011, 2013, Hansen et al. 2011, Marshall et al. 2014, Ferreira et al. 2015). As discussed in Hasselmann et al. (1993), for example, step function response experiments have a long history in climate science and are related to 'impulse' (Green's) function responses. Here we propose a coordinated program of research in which we compute CRFs for the Arctic in response to key Arctic 'switches', as indicated schematically in Fig.1.

A successful coordinated activity has a low bar for entry, is straightforward to carry out, involves models of all kinds — low resolution, high resolution, coupled and ocean only — is exciting and interesting scientifically, connects to observations and, particularly in the context of the Arctic, to climate change and the climate change community. Our hope is that the activity set out here satisfies many of these goals. The ideas were presented to the FAMOS (Forum for Arctic Modeling & Observational Synthesis[1] community in the Fall of 2016. This paper stems from those discussions and sets out in a more formalized way how

---

[1]see http://famosarctic.com

to compute CRFs for the Arctic, what they might look like, and proposed usage. We invite Arctic modelers and observers to get involved.

The main "switches" for the Arctic Ocean are, as indicated schematically in Fig.1:

1. Wind forcing — increasing and decreasing the wind field both within the Arctic basin ($W_I$) and (just) outside the basin ($W_O$).

2. Freshwater forcing — stepping up and down the river ($R$) and ($E - P$) freshwater fluxes.

3. Inflows — changes in the heat and freshwater flux, either by volume, or inflow temperature/salinity from the Atlantic ($A$) and Pacific ($B$) of water flowing in to the Arctic Ocean.

Each participating group would choose their preferred Arctic simulation and perturb it with exactly the same forcing fields in exactly the same manner. All other modelling choices would be left to the discretion of the individual groups. Suggested forms for, and examples of $W_I$, $W_O$, $R$, $E - P$, $A$ and $B$ are discussed and described here. 'Observables', such as the freshwater content of the Beaufort Gyre, sea-ice extent etc., would be computed, evolution maps and time-series plotted and compared across the models. Differences/similarities across models will motivate scientific discussion. Convolutions with observed time-series of the forcing (an example is given Section 3.5) allow comparisons to be made with observations (retrospectively) and climate change projections from models.

Our paper is set out as follows. In Section 2 we motivate how we propose to compute CRFs for key observables and forcing functions in the Arctic. In Section 3 we illustrate the approach in the context of a coarse-resolution model of the Arctic based on the MITgcm. There we compute CRFs for the 'switches' shown in Fig.1 and demonstrate how convolutions can be computed for a chosen time-series of the forcing from knowledge of the model response to a step. In Section 4 we outline a suggested protocol enabling other groups to carry out the same experiments. We conclude in Section 5 with a summary of expected benefits.

## 2    Motivation behind Arctic perturbation experiments

### 2.1    Response to step-functions in the forcing

Much community effort goes in to building and tuning models of the Arctic that have the best possible climatology and seasonal cycle, as measured against observations. Previous coordinated experiments have compared the climate states of these models and their sensitivity to parameters and forcing fields (see, e.g. Proshutinsky et al. 2011; Ilicak et al. 2016). But one is also keenly interested in how the system responds to a *change* in the forcing, as, e.g., in the idealized study of Lique et al. 2015. This is perhaps particularly true in the Arctic which is undergoing rapid change as the Earth warms. Indeed much of climate research focuses on the change under anthropogenic forcing, rather than the mean climate. Of course fidelity in the mean might be a prerequisite for fidelity in the forced response, but this is not always the case. For example, one can make a rather good prediction of the change of global mean SST with a simple (albeit tuned) 1-d energy balance model which makes no attempt

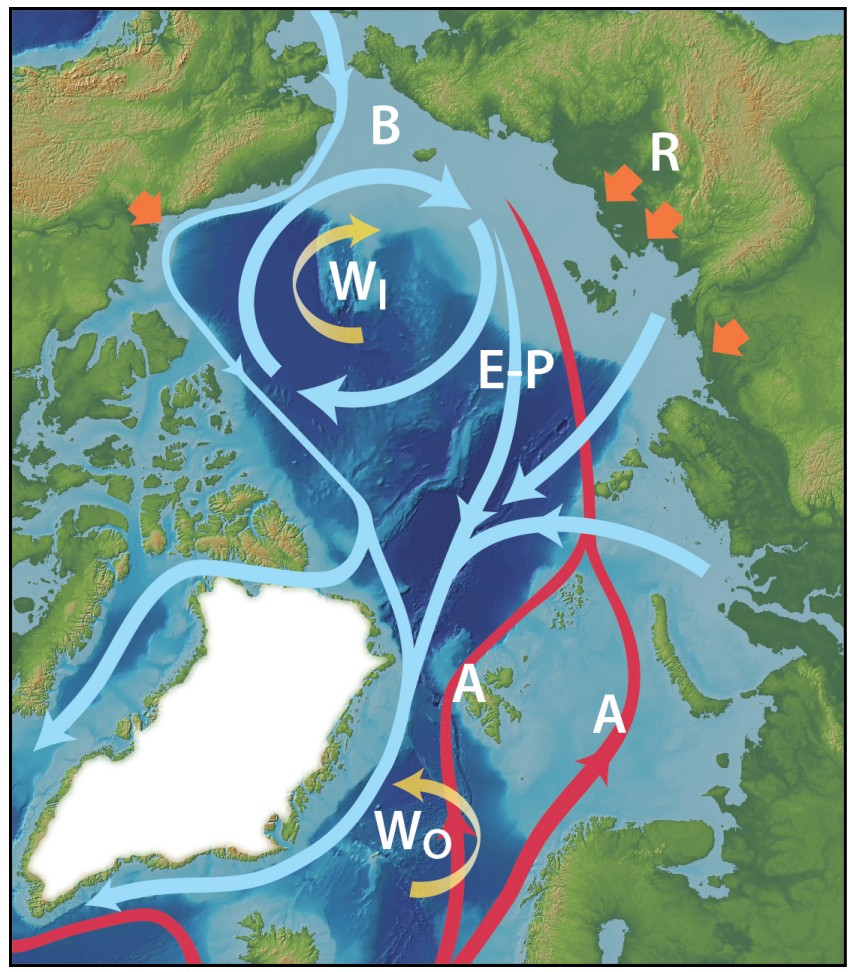

**Figure 1.** A schematic of circulation pathways in the Arctic Ocean and key 'switches' that can perturb it. Background colour coding is ocean bathymetry and elevation over land. Thick blue pathways show general branches of sea ice drift and surface water circulation. "$B$" indicates the entrance of Pacific waters to the Arctic Ocean through the Bering Strait. The thin blue pathways originating in the Bering Strait region depict a hypothetical branch of Pacific water flow involved in the coastal boundary current. Red arrows represent inflows of warm Atlantic waters entering the Arctic Ocean via Fram Strait and through the northern parts of the Kara Sea. Note that in the Fram Strait region and the Barents Sea, these branches of Atlantic water (depicted as "$A$") enter the Arctic Ocean and subsequently circulate around it at depths greater than $100 - 150$ meters. Key 'switches' for the Arctic, which will be perturbed in our models, are also indicated: winds interior ($W_I$ in the Beaufort Gyre) and exterior ($W_O$ in the Greenland Gyre) to the Arctic basin, river runoff ($R$, orange arrows), evaporation/precipitation ($E - P$) and inflow of Atlantic ($A$) and Pacific (through the Bering Strait region $B$).

to capture three-dimensional dynamics. Much of the IPCC process concerns comparing changes in model states under forcing rather the mean states of those models.

The coordinated experiments we are proposing here focus on the response of Arctic models to external forcing rather than comparing mean states. We organise our discussion around 'Climate Response Functions' (CRFs) i.e. the response of the Arctic to 'step' changes in forcing, as represented schematically in Fig.1, and the transient response of the system is revealed and studied.

### Why step-functions?

Step functions have a special status because they are the integral in time of the impulse response from which, in principle, one can construct the linear response to any time-history of the forcing: if one knows the CRF and the respective forcing function, convolving one with the other yields the predicted linear response (see, e.g. Section 3.5).

More precisely we may write, (see, e.g. Marshall et al. 2014):

$$\mathcal{R}(t) = \int\limits_0^t CRF(t-t')\frac{\partial F}{\partial t}(t')dt', \tag{1}$$

where $F$ is the prescribed forcing function (in HPa for a pressure perturbation producing anomalous winds)[2], $CRF$ is the step response function per unit forcing and $\mathcal{R}(t)$ is the response. For example, $\mathcal{R}$ might be summertime Arctic sea-ice extent, $F$ the wind field over the Beaufort Gyre and $CRF$ the response function of the ice to the wind. Many observables could be chosen depending on the question under study and the availability of observational time-series. But it is important that they be chosen with care and represent some integral measure of Arctic response.

The "magic", then, is that if we know the response function of a diagnostic quantity to a step change in a chosen forcing, we can then convolve this response function with a time-history of the forcing to obtain a prediction of the linear response to that forcing history, without having to run the actual experiment. This can be checked *a-posteriori* by running the true experiment and comparing the predicted response to the convolution, as given in Section 3.5.

Finally, more support for the idea of computing the step response comes from Good et al. (2011, 2013) in which the response of climate models to abrupt $4\times CO_2$ is used to predict global mean temperature change and ocean heat uptake under scenarios that had not been run. Gregory et al. (2015) shows how the step approach is a good way to distinguish linear and non-linear response in global predictions. In the same way, our project will be able to ascertain the degree of linearity of Arctic CRFs. It should be emphasised that if the system is not linear, convolutions would then provide only limited predictive skill. This may be true of, for example, Arctic sea ice cover, given the strongly nonlinear nature of ice. One might also expect the linear assumptions to break down as the amplitude of the forcing is increased, a point to which we return below.

---

[2]or Sv for freshwater forcing, or PW for the heat flux anomaly associated with Arctic inflow etc.

## 2.2 Choosing key Arctic forcing functions and observables

### 2.2.1 Forcing functions

The key switches for the Arctic Ocean are set out schematically in Fig.1 and comprise wind anomalies both interior ($W_I$) to the Arctic and exterior to it ($W_O$), perturbations to the runoff ($R$) and ocean transports into the Arctic from outside ($A$ and $B$). To illustrate our approach here we focus on perturbations to the wind field over the Beaufort Gyre and the Greenland Sea, the heat flux through Fram Strait and river runoff. Many other perturbations could also be considered. Our choice of switches are motivated by the following considerations.

Wind forcing: Wind is one of the most important forcing parameters driving variability of ice drift and ocean circulation ('wind blows, ice goes', a rule of thumb well known since Arctic exploration in the 17th century) and responsible for mechanical changes in sea ice concentration and thickness, freshwater content variability, upwelling and downwelling processes with implications for both oceanic geochemistry and ecosystem changes.

There are two major wind-driven circulation regimes over the Arctic Ocean, namely: cyclonic and anticyclonic having decadal variability with significant differences in environmental parameters between these regimes (Proshutinsky and Johnson, 1997; Proshutinsky et al., 2002; Thompson and Wallace, 1998; Rigor et al., 2001; Proshutinsky et al., 2015). The Beaufort Gyre and Greenland Gyre regions are key circulation cells in the central Arctic Ocean and central Nordic seas and regulated by Beaufort and Icelandic High atmospheric systems, respectively. In our recommended experiments, anomalous wind direction and intensity in these regions have been chosen based on observational data (NCAR/NCEP reanalysis products).

River runoff: River runoff is the major source of freshwater for the Arctic Ocean. The freshwater is a key component in the Arctic hydrological cycle affecting ocean, sea ice and atmosphere. In the Arctic Ocean, the FW at the surface maintains a strong stratification that prevents release of significant deep-ocean heat to the sea ice and atmosphere (i.e. halocline catastrophe, Aagaard and Carmack, 1989; Toole et al., 2010).

Arctic FW exports can affect climate of the North Atlantic by potentially disrupting deep convection in the North Atlantic, and it can affect the Atlantic Meridional Overturning Circulation (AMOC) if Arctic fresh water reaches convective sites in the Labrador Sea (Yang et al., 2016), for example. Thus, understanding the response to river runoff (especially as the climate warms and the hydrological cycle intensifies) is important for predicting future change. Numelinn et al. (2016) and Pemberton and Nilsson (2016), for example, have found that increased river runoff leads to a strengthening of the central Arctic Ocean stratification and a warming of the halocline and Atlantic Water layers. Further, excess fresh water accumulates in the Eurasian Basin, resulting in local sea level rise and a reduction of water exchange between the Arctic Ocean and the North Pacific and North Atlantic Oceans. Thus we expect our recommended experiments, with different scenarios of runoff intensity and employing a set of models with different resolutions and parameterisations, will shed light on these problems.

Fram Strait salt and heat fluxes: One of the fundamental aspects of the Arctic Ocean is the circulation and transformation of Atlantic Water, which plays a critical role in Earth's climate system. Profound modification and conversion of these waters into North Atlantic Deep Water occur within the Arctic, making this region the 'headwaters' of the global meridional overturning circulation. As far back as the early 1900s Nansen concluded that the warm intermediate layer of the Arctic Ocean originated

in the North Atlantic Ocean; oceanographers have since explored the intricate pathways, behaviour, and impacts of Atlantic waters throughout the Arctic basin. While we have gained an understanding and appreciation of the importance of Atlantic Water, much remains to be learned. In our recommended experiments, the heat flux through the Fram Strait is perturbed, enabling us to test a set of hypotheses about the role Atlantic waters play in the Arctic. One of them is that heat release from the Atlantic water layer is responsible for sea ice decline in the Arctic Ocean (e.g. Carmack et al., et al., 2015). CRF experiments will also shed light on the pathways and intensity of Atlantic water and the interaction of boundary currents with the Arctic interior.

### 2.2.2 Observables

Ideal observables — the left-hand side of Eq.(1) — are integrated quantities (not, for example, the temperature at one point in space), which should be constrained by observations, indicative of underlying mechanisms and of climatic relevance. Two key attributes of useful 'observables' are worth emphasising: a) those that make reference to existing theories/hypotheses about Arctic ocean dynamics; their CRFS can then inform our understanding and b) those for which CRFs can be constructed from observations, providing a quantitative measure for evaluation of model skill. With regard to the latter, given the difficulty of obtaining in-situ observations, our focus is on large-scale integrated quantities. Some of the best available are satellite-derived, e.g. sea ice concentration (and ice area and extent derived from it) and ice drift from CryoSat, freshwater content inferred from CryoSat's sea-surface height fields and sea-surface temperatures in open water areas. Ocean fluxes through Straits are perhaps best constrained by in-situ observations, although they suffer from a lack of near-surface observations (i.e. Woodgate et al., 2015; Beszczynska-Möller et al., 2012), especially for the freshwater flux.

The following Arctic 'observables/metrics' are a useful starting point, each one of which is constrained to some degree by observations:

- Freshwater and heat storage of the Beaufort Gyre,

- Strength of boundary currents,

- Summer and winter sea-ice extent, sea-ice thickness and volume,

- Flux through various sections and Straits,

- Mixed layer depth,

- Export of heat and freshwater to the North Atlantic Ocean.

  Some of the key regions and sections that are of interest to us are shown in Fig.2. Many others could also and are being considered.

### 2.3 Science questions

Key science questions are:

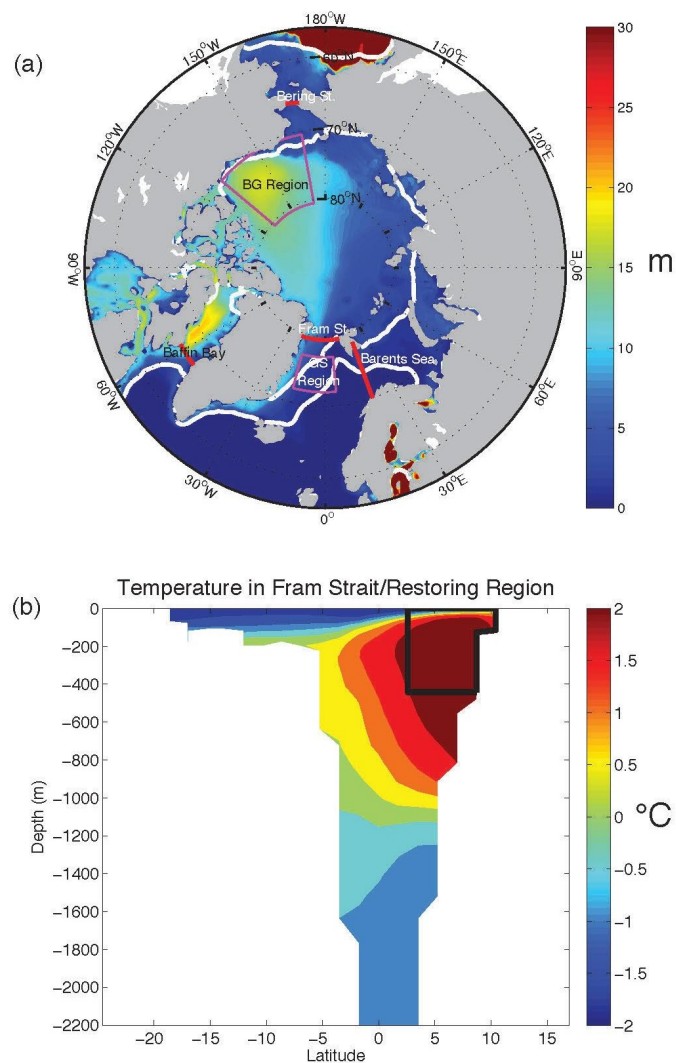

**Figure 2.** (a) Average FWC (fresh-water content) over the period 1979-2013 (coloured in m) from the MITgcm simulation. The summer (inner white lines) and winter sea-ice extent (outer white lines) are plotted. Key sections and regions are labelled. (b) Annual-mean temperature section through the Fram Strait looking northward in to the Arctic. The black box indicates the region where inflow parameters are modified in the calculations presented.

- What sets the time-scale of response of the above metrics to abrupt changes in the forcing? Some will respond rapidly to changes in forcing, others more slowly. Can we understand why in terms of controlling physical processes?

- Are responses symmetric with respect to the sign of the perturbation? This may simply not be true in the presence of sea ice when on-off behaviour can be expected. Moreover, linearity is likely to be a function of the magnitude of the applied perturbation and will likely break down if the perturbation is too large.

- Do some observables exhibit threshold behaviour, or indicate the possibility of hysteresis?

- Do convolutions of the observed forcing with the CRF shed light on observed time-series?

We do not have space to explore all these issues here but return to some of them in the conclusions. We now go on to present examples of the experiments we are proposing.

## 3  Illustrative examples with a 'realistic' Arctic ocean model

To give a concrete example of Arctic CRFs, in this section we compute the response of a coarse-resolution model of the Arctic based on the MITgcm (Marshall et al, 1997a,b; Adcroft et al. 1997) to step changes of the forcing shown in Fig.1. We first describe the climatology of the model, the forcing functions that we use to perturb it, describe the resulting CRFs and show that they can be used to reconstruct the model's response to a time-dependent forcing.

### 3.1  Arctic model based on the MITgcm

#### 3.1.1  Configuration

The simulation is integrated on the Arctic cap of a cubed-sphere grid, permitting relatively even grid spacing throughout the domain and avoiding polar singularities (Adcroft et al., 2004). The Arctic face comprises 210 by 192 grid cells with a mean horizontal grid spacing of 36km. A linearised free surface is employed. There are 50 vertical levels ranging in thickness from 10m near the surface to approximately 450m at a maximum model depth of 6150m. Bathymetry is from the 2004 (W. Smith, unpublished) blend of the Smith and Sandwell (1997) and the General Bathymetric Charts of the Oceans (GEBCO) one arc-minute bathymetric grid. The non-linear equation of state of Jackett and McDougall (1995) is used. Vertical mixing follows Large et al. (1994) with a background diffusivity of $5.4 \times 10^{-7}$ m$^2$s$^{-1}$. A 7th-order monotonicity-preserving advection scheme (Daru and Tenaud, 2004) is employed and there is no explicit horizontal diffusivity. A mesoscale eddy parameterisation in the spirit of Gent and McWilliams (1990) is used with the eddy diffusivity set to $K = 50$m$^2$s$^{-1}$. The ocean model is coupled to a sea ice model described in Losche et al. (2010) and Heimbach at al. (2010).

The 36km resolution model was forced by the JRA-25 (6hr, 1°) reanalysis for the period 1979-2013, using bulk formulae following Large and Pond (1981). Initial conditions for the ocean is the WOCE Global Hydrographic Climatology (annual-mean, 1990-1998 from Gouretski and Koltermann, 2004). Open boundary conditions on $S, T, u$ & $v$ were employed using

'normal-year' conditions averaged from 1992–2002 derived from an ECCO climatology (Nguyen, Menemenlis and Kwok, 2011). Decadal runs take a few hours on 80 cores.[3]

### 3.1.2 Climatology

Our model has a reasonable climatology, as briefly illustrated in Fig.2 and Fig.3. Fig.2a shows a plan view of the FWC (freshwater content, see Aagaard and Carmack, 1989)[4] in the BG averaged over the period 1979-2013. It has a plausible structure and is broadly in accord with, e.g., Fig.6 of Haine et al. 2015, both in magnitude and spatial pattern. The winter ice-edge is marked by the 'outer' white lines, the summer ice edge by the 'inner' lines. Comparison with observations reveals that our modelled sea-ice is rather too extensive, both at the summertime minimum and the wintertime maximum.

Time-series of FWC and HC (top 400m) over the Beaufort Gyre (the horizontal region over which we integrate is delineated by the box in Fig. 2a) is shown in Fig.3a. Fig. 3a reveals that the freshwater and heat content are varying on decadal timescales with an increased accumulation of FWC[5] (by roughly $2500 \mathrm{km}^3$) in the 2000s and a leveling out in heat content relative to earlier decades. The recent trends (of order 10% of the mean) may have been associated with an increased anticyclonic wind over the BG (Proshutinsky et al. 2009; Rabe et al. 2014). The evidence is reviewed in Haine et al. (2015).

It is also clear from Fig.3 that the model is drifting with a downward/upward trend in FWC/HC. The model described here has undergone no data-assimilative procedure and so might be expected to exhibit such drifts as it adjusts to the initial conditions and forcing.

Figs.3c plots the annual cycle of sea-ice area from the 1980s onwards showing a decline in the minimum (summer) ice area of order $10^6 \mathrm{km}^2$ in 30 years. The observed rate of sea ice extent loss is much more dramatic than captured in our model: observations suggest that sea-ice has declined by $\sim 0.5 \times 10^6 \mathrm{km}^2$ per decade (annual mean) in the last few decades to below $8 \times 10^6 \mathrm{km}^2$ (see, e.g., Fig.1a of Proshutinsky et al. 2015) whereas the modelled annual-mean area is $11 \times 10^6 \mathrm{km}^2$.

Fig.2b shows a vertical temperature section through our model, roughly coinciding with Fram Strait (as indicated in Fig.2a), and Fig.3e-f plots time-series of FWF (freshwater flux) and HF (heat flux) through the Strait: positive indicates a flux into the Arctic, negative out of the Arctic. We observe a strong seasonal cycle and much interannual variability superimposed on longer-term trends/drifts. The magnitude of the mean value of FWF is somewhat smaller than the $2700 \pm 530 \mathrm{km}^3 \mathrm{y}^{-1}$ estimated from observations (see Table 1 and Fig.4 of Haine et al. 2015). The HF through Fram Strait varies by $\sim 10 \mathrm{TW}$ over the period of our simulation, roughly comparable with the CORE ocean models reported in Ilicak et al. (2016).

---

[3]Very similar 18km and 4km configurations of the same model exist and can be used in eddy permitting and resolving simulations for comparison with the parameterised model.

[4] Freshwater content is defined here (as reviewed in Haine et al, 2015), as the amount of zero-salinity water required to reach the observed salinity in a seawater sample starting from a reference salinity. It is computed as: $FWC = \int_{D}^{\eta} \frac{S_{ref}-S}{S_{ref}} dz$ where $\eta$ is the free surface and we choose $S_{ref} = 34.80$ and $D$ is its depth. This is the quantity mapped in Fig.2a. Similarly, freshwater flux (FWF) is defined by multiplying the integrand of the above expression by velocity and integrating along the section.

[5] To convert the FWC of the BG to meters of freshwater, divide by the surface area of the BG, here taken to be $1.24 \times 10^6 \mathrm{km}^2$, the area of the box in Fig.2a. Thus a FWC $= 20 \times 10^3 \mathrm{km}^3$ corresponds to a depth of $\frac{20 \times 10^3 \mathrm{km}^3}{1.24 \times 10^6 \mathrm{km}^2} = 16 \mathrm{m}$ of fresh water, roughly in accord with observations — see, e.g., Fig.6 of Haine et al. (2015) and Fig. 2a.

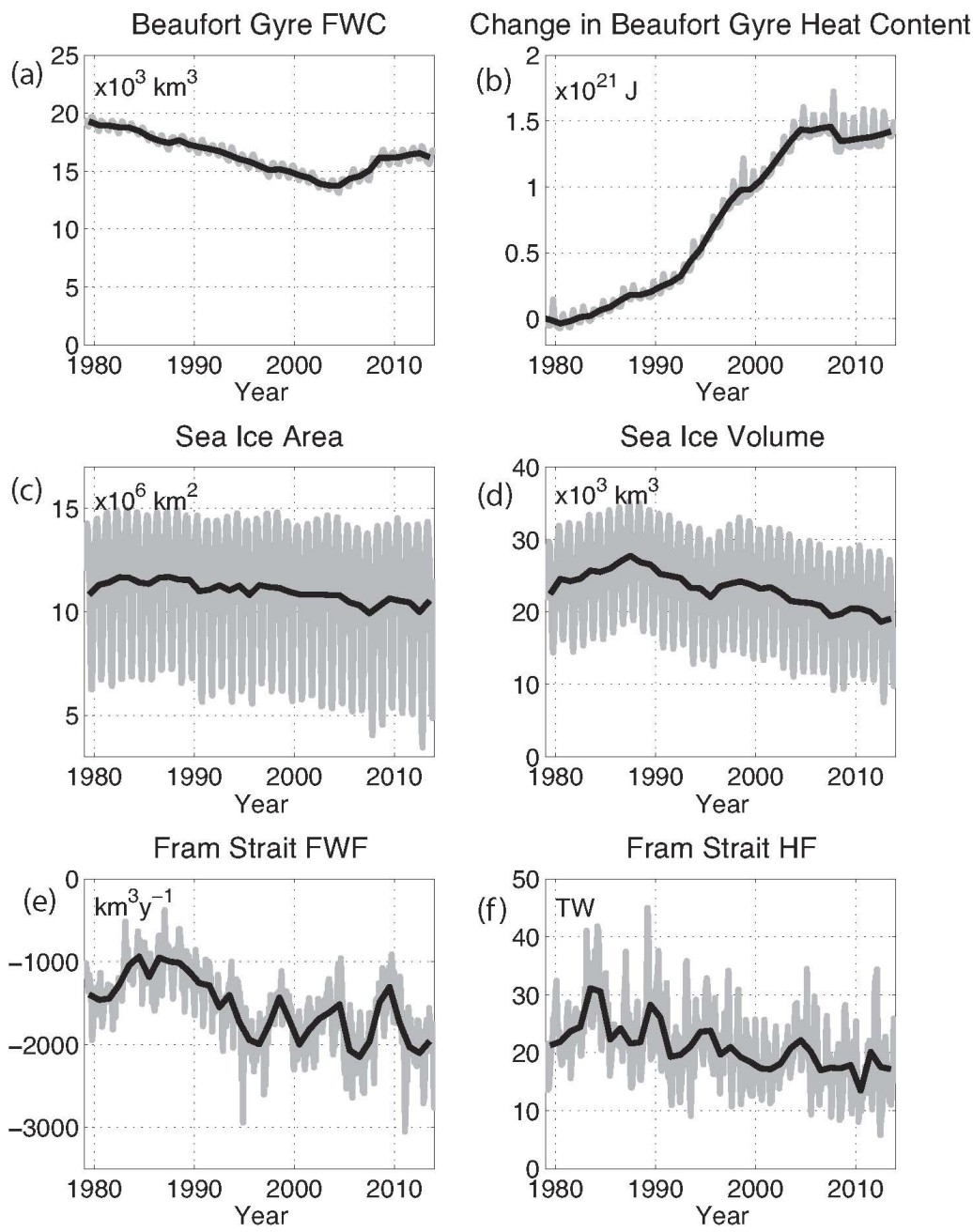

**Figure 3.** Time-series of (a) fresh water content (FWC) and (b) heat content (HC) of the BG, (c) sea-ice area and (d) sea-ice volume over the Arctic, (e) fresh water flux (FWF, negative values imply a flux out of the Arctic) and (f) heat flux through Fram Strait (HF, positive values indicate a flux in to the Arctic). The thick black line plots annual-mean values, the grey line tracks monthly-means.

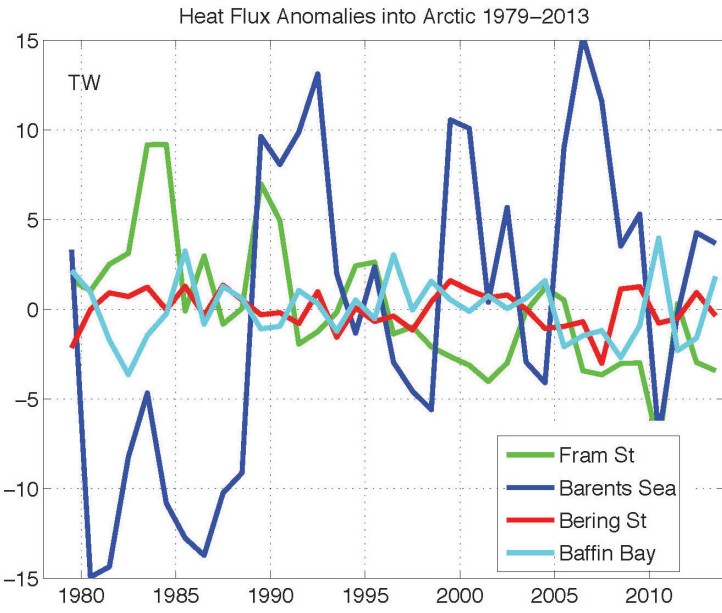

**Figure 4.** Heat flux anomalies (seasonal cyle removed) across key Arctic Straits as indicated in Fig.2. The units are in TW.

In Fig.4 we plot time series of annual-mean anomalous heat flux through various Arctic Straits shown in Fig.2a. We observe, for example, that heat transport through the Barents Sea Strait is increasing and that through Fram Strait is decreasing with a strong decadal trend. In contrast the transport through the Bering Strait and Baffin Bay vary primarily on internannual timescales with less evidence of decadal trends. Comparison of the timeseries shown in Fig.4 with those in Figs.11 through 14 of Ilicak et al. (2016) shows broad similarities despite the fact that the latter study uses CORE forcing and a variety of models employing differing physical parameterisations, open boundary conditions and grid resolutions.

It is clear from the above brief review of key circulation and sea-ice metrics (clearly many more are likely to be of interest) that they respond to the various external drivers in different ways with respect to amplitude and timescale. As we now go on to describe, we can expose and explore some of the underlying mechanisms by computing how the model responds to a step increase in the forcing.

### 3.2 Anomalies in forcing functions

We now describe the prescription of the forcing function anomalies in wind, river runoff and transport through Straits.

### 3.2.1 Wind

Simplified forms of the surface pressure anomalies over the Beaufort Gyre (BG) and Greenland Sea (GS) have been constructed and are plotted in Fig.5. The centre for the BG pressure anomaly is located at (77°, 149°W) and the centre for the GS anomaly is located at (71°, 6°W), with a radius of influence on the order of 1000km. The magnitude of the anomaly is the same for all experiments. Our choice of BG and GS atmospheric centres of action were identified based on 1948—2015 6-hourly NCAR/NCEP data. These data were analysed to identify key locations of centres of action and typical magnitudes north of the Arctic Circle. These centres can also be determined from Thompson and Wallace's (1998, 2001) studies of the Arctic Oscillation (AO, first mode of SLP EOF analysis which describes approximately 19% of SLP variability in December – March).

In the series of experiments described here we assumed a central maximum/minimum of 4HPa. Our perturbation of 4HPa is small relative to seasonal changes, which can reach 20-30 HPa. However, a more reasonable measure is to compare with longer-term trends. During the 1948-2015 period, SLP over the Arctic changed by about 6 HPa suggesting that our chosen magnitude is not unrealistic. As can be seen by inspection of the right hand panels of Fig.5, there is a noticeable perturbation to the long-term climatology of SLP when anomalies of this magnitude are assumed.

To compute surface winds from these pressure anomalies, the following relation is used (Proshutinsky and Johnson, 1997):

$$W_s = 0.7 \times \begin{bmatrix} \cos 30 & -\sin 30 \\ \sin 30 & \cos 30 \end{bmatrix} \times W_g$$

where $W_g$ is geostrophic wind implied by the pressure anomaly, and $W_s$ is the applied surface wind anomaly. The resulting anomalous winds are also plotted in Fig.5.

### 3.2.2 Fluxes through Straits

We perturb the properties of water masses flowing through Fram Strait (FRAM). For simplicity we aligned the section with our model grid, extending from gridpoints centred at (80°, 10°E) [near Svalbard] to (80°, 19°W) [the Greenland coast], marking a line close to a true parallel (see Fig. 2a). Our objective is to perturb the temperature of water flowing across the section into the Arctic, but without a concomitant density change. This is accomplished by rapid restoring of temperature while simultaneously restoring salt to compensate. In the experiments described here, the restoring temperature was T+2K and restoring salinity was S+0.253 psu[6] where both T and S were monthly fields diagnosed from our 35-year control run. The restoring area was limited both in zonal extent and depth along the section: from (80°, 10°E) [Svalbard coast] to (80°, 3.5°E), in the vertical from the surface to 440m, as indicated by the box in Fig.2b. The box was chosen to capture the main core of Atlantic water entering the Arctic through the Strait. A restoring time constant of 9 days was used for both T and S. Finally, for numerical reasons we opted to also restore, to the same T+2K and S+.253 psu anomalies, adjacent gridpoints both to the north and south of the aforementioned section, on a longer timescale of 90 days. This simple procedure ensures that the potential density in the Fram section in the control and the forced experiment are very similar.

---

[6]Compensating salinity, $\delta S = \frac{\alpha}{\beta} \delta T = 0.253$psu, was computed using a $\delta T = 2$K assuming $\alpha = 1 \times 10^{-4} \mathrm{K}^{-1}$, $\beta = 7.9 \times 10^{-4} \mathrm{psu}^{-1}$, roughly corresponding to 5K, 33psu seawater.

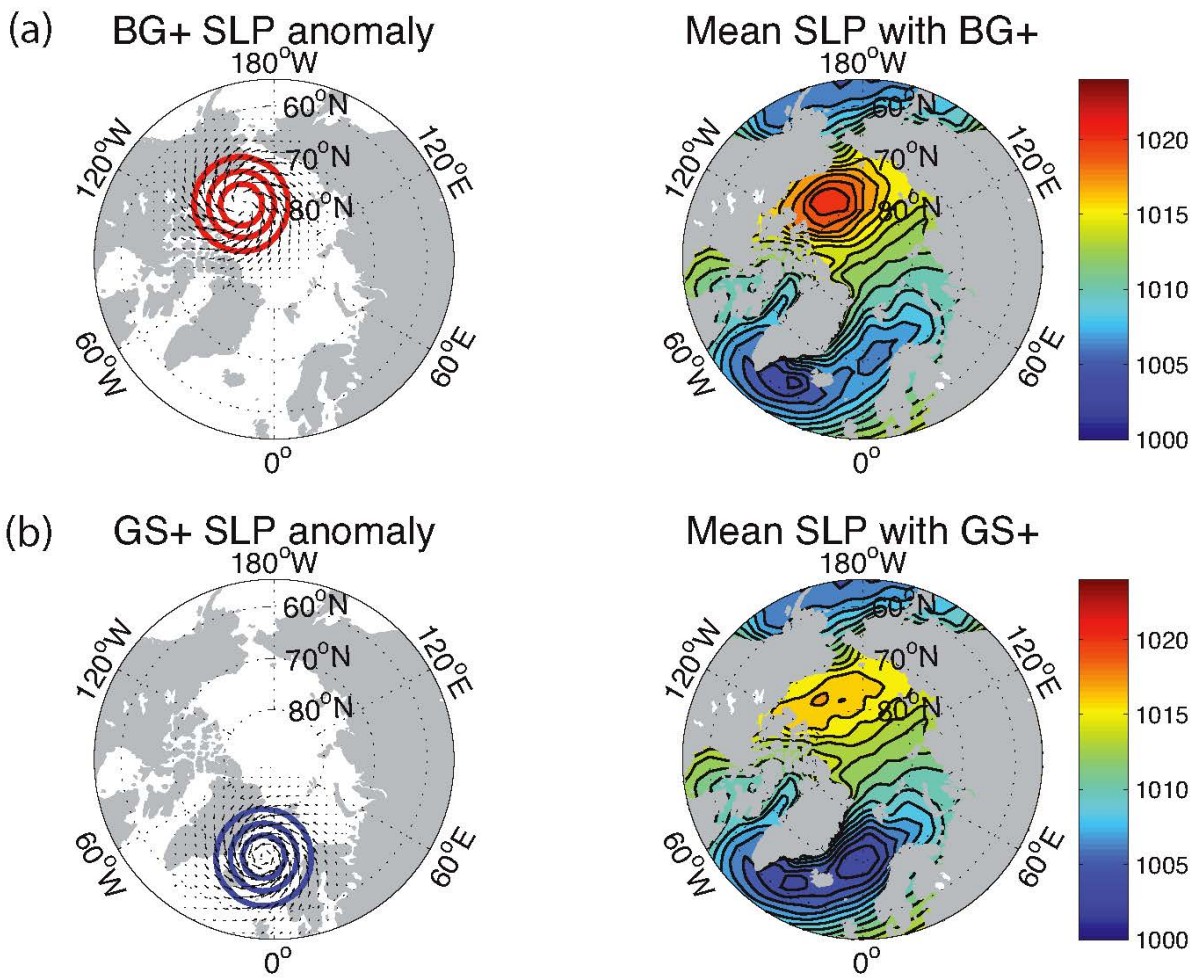

**Figure 5.** (a) (left) Idealized sea-level pressure anomaly of 4HPa and associated winds constructed for the Beaufort Gyre (BG). Note the BG(+) corresponds to anomalously high pressure. (right) BG(+) SLP anomaly added to the NCEP 1981-2010 SLP climatology. (b) (left) Idealized sea-level pressure anomaly of 4HPa and associated winds constructed for the Greenland Sea (GS). Note the GS(+) corresponds to anomalously low pressure over the GS. (right) GS SLP anomaly added to the 1981—2010 SLP climatology.

### 3.2.3 Runoff

For the anomalous river runoff experiment (RUN3x), the freshwater input from all rivers which drain into the ocean north of the Arctic Circle (66 ° N) was multiplied by a factor of three. In our regional Arctic setup, no effort was made to balance this anomalous fresh water input with additional evaporation.

### 3.3 Climate Response Functions

Figs.6, 7 & 8 show the CRFs for, respectively, the BG wind anomaly, the GS wind anomaly and runoff/Fram Strait temperature anomaly. The forcing anomalies are applied one-at-a-time, although combinations would also be of interest. We focus on metrics of FWC, HC, sea-ice area and volume and Fram Strait FWF and HF. This is an interesting subset of the large number of circulation and ice metrics that could be computed and discussed. There are interesting spatial patterns of response but they are not discussed here.

In Fig.6 the CRFs of key quantities for the (+) and (-) BG wind anomalies are shown. The (+) sign indicates that the Beaufort High is anomalously strong with enhanced anticyclonic flow. We see that in response to anomalously high surface pressure over the BG, its FWC increases, readjusting to a new quasi-equilibrium after about 30 years but continuing to trend upward. An increase in FWC is to be expected since enhanced anticyclonic winds and their associated Ekman transport draw fresh water from the periphery of the BG, increasing its FWC. As the BG freshens it also becomes colder, as seen by its decreasing heat content (Fig.6b). Thus freshwater and temperature changes tend to compensate one-another with respect to their effect on density. We see from Fig.6c, there is little change in the sea-ice area in response to the enhancement of the anticyclonic wind field, but a substantial increase in the volume of sea-ice: evidently sea-ice is converging and thickening.

In Fig.7 the CRFs of key quantities for the (+) and (-) GS wind anomalies are shown. Note that (+) indicates that the low pressure system that resides over the GS (the northward extension of the Icelandic Low) is anomalously strong, thus inducing anomalously cyclonic circulation over the Barents and Greenland Seas — see Fig.5b. In response to GS(+)/GS(-) the BG FWC increases/decreases slightly, but with a delay of 10 years or so. This is presumably an advective signal. There is a pronounced change (but again with a decadal delay) in the HC of the BC, which warms in the (-) case and cools in the (+) case. Unlike for the BG wind forcing, we observe a notable increase in sea-ice area for a (-) anomaly and a decrease for a (+) anomaly. An increase in low pressure over the GS leads to increased advection of sea-ice out of the Arctic through Fram Strait and advection of warm water into the Arctic resulting in ice melt: both factors lead to a decrease in sea-ice area. Changes in sea-ice volume are also observed but with reduced magnitude relative to the BG wind anomalies. CRFs for Fram Strait FWF and HF induced by GS wind anomalies are all suggestive of a two timescale process at work — with the response changing sign after 10 years or so in the case of the Fram FWF and after 5 years or so in the case of the Fram HF.

Fig.8 shows the response of our key metrics to changes in runoff and Fram Strait heat transport implemented as described in Sections 3.2.2 and 3.2.3. It should be noted that these are rather large perturbations, much greater than might be expected to occur naturally. We see that as runoff is increased, the southward FWF through Fram Strait increases linearly over a 30 year period with a compensating northward heat flux, the FWC of the BG increases very slightly, as does sea-ice area and volume.

An impulsive increase in the HF through Fram Strait leads to an increase in the HC of the BG after a decade or so but has no discernible effect on the other metrics under consideration.

Some of our results can be compared with findings of Nummelin et al. (2015, 2016) and Pemberton and Nilsson (2016) who studied the impact of river discharge on the Arctic Ocean. Both studies assumed that future Arctic river runoff will likely increase due to intensification of the global hydrological cycle. One interesting finding of the latter study was that under an increased freshwater supply, the Beaufort Gyre weakens and there is increased freshwater exported through Fram Strait. Here, FWC of the BG is indeed insensitive to runoff (Fig.8a) and instead results in increased freshwater flux through Fram Strait (Fig.8e). Narrow fresh coastal flows can explain the insensitivity of BG FWC to the increased river runoff. Evidently most of the freshwater is transported directly to the Fram and Canadian Straits rather than being accumulated in the BG region.

In summary, the following general features of the CRFs are worth noting:

1. The CRFs do not respond immediately to a step in the forcing, but adjust over time, on a timescale that depends on the metric and the forcing being considered.

2. Some CRFs (e.g. FWC) have a simple form that can be characterised by a single timescale. Others are suggestive of a two timescales and/or zero-crossing (change of sign) behaviour (eg. Fram Strait HTF and FWF).

3. The CRFs are (roughly, but not all) symmetric with respect to a change in the sign of the forcing, as one would expect if the behaviour were linear. Note, however, that as the amplitude of the forcing is increased to rather unrealistic levels, asymmetries in the response become more prevalent (not discussed further here).

### 3.4 Ensembles

To test the robustness of our CRFs we generate an ensemble by varying the time of onset of the forcing step function. In Fig.9, we show the CRFs for (a) the FWC in the Beaufort Gyre (b) and the heat transport through Fram Strait, varying the start time of the BG+ wind anomaly step function to day 1 of each month over the run's first year. We see that the FWC CRF shows minimal impact to varying the seasonal timing of the forcing anomaly. This is not surprising given that FWC is an integrated quantity over the upper ocean salinity field. On the other hand, the heat flux through Fram Strait shows a much larger envelope in the collective ensemble CRF, yet maintains the same basic form. It will be useful to compare similarly generated ensembles across other models for these and other model metrics. Our calculations suggest that not many ensemble members — perhaps 5 — will be required, at least in coarse resolution models such as the one used here.

### 3.5 Convolutions

Having described the form of some key CRFs, we now convolve them with periodic forcing functions to compute the implied linear response of, for example, an oscillating wind anomaly. This is then compared to direct calculations with our full ocean model to provide a sanity check on our methodology and the utility of CRFs. To make things concrete we will focus on the FWC of the BG and wind anomalies over the BG.

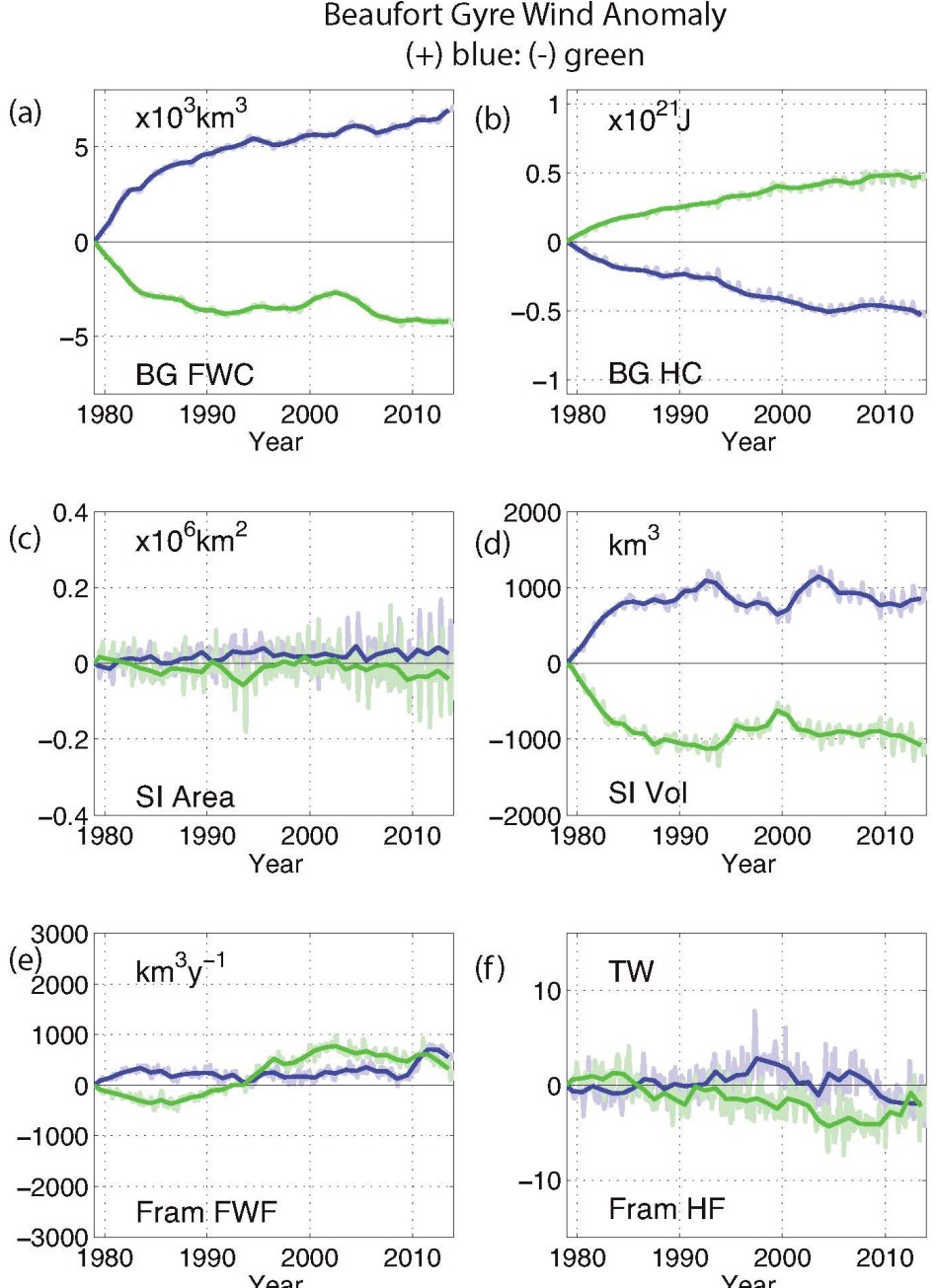

**Figure 6.** CRFs for the Beaufort Gyre wind anomaly, blue (+) and green (-). Note that the (+) sign implies a stronger anti-cyclonic forcing. The response to a 4HPa surface pressure anomaly (see Fig.5a) is shown of (a) $C^{W_{BG}}_{FWC_{BG}}$, the FWC of the BG (b) $C^{W_{BG}}_{HC_{BG}}$, HC of the BG (c) $C^{W_{BG}}_{SIA}$ , SI area (d) $C^{W_{BG}}_{SIV}$, SI volume (e) $C^{W_{BG}}_{FWF_{Fram}}$, FWF through the Fram Strait and (f) $C^{W_{BG}}_{HF_{Fram}}$, the HF through the Fram Strait.

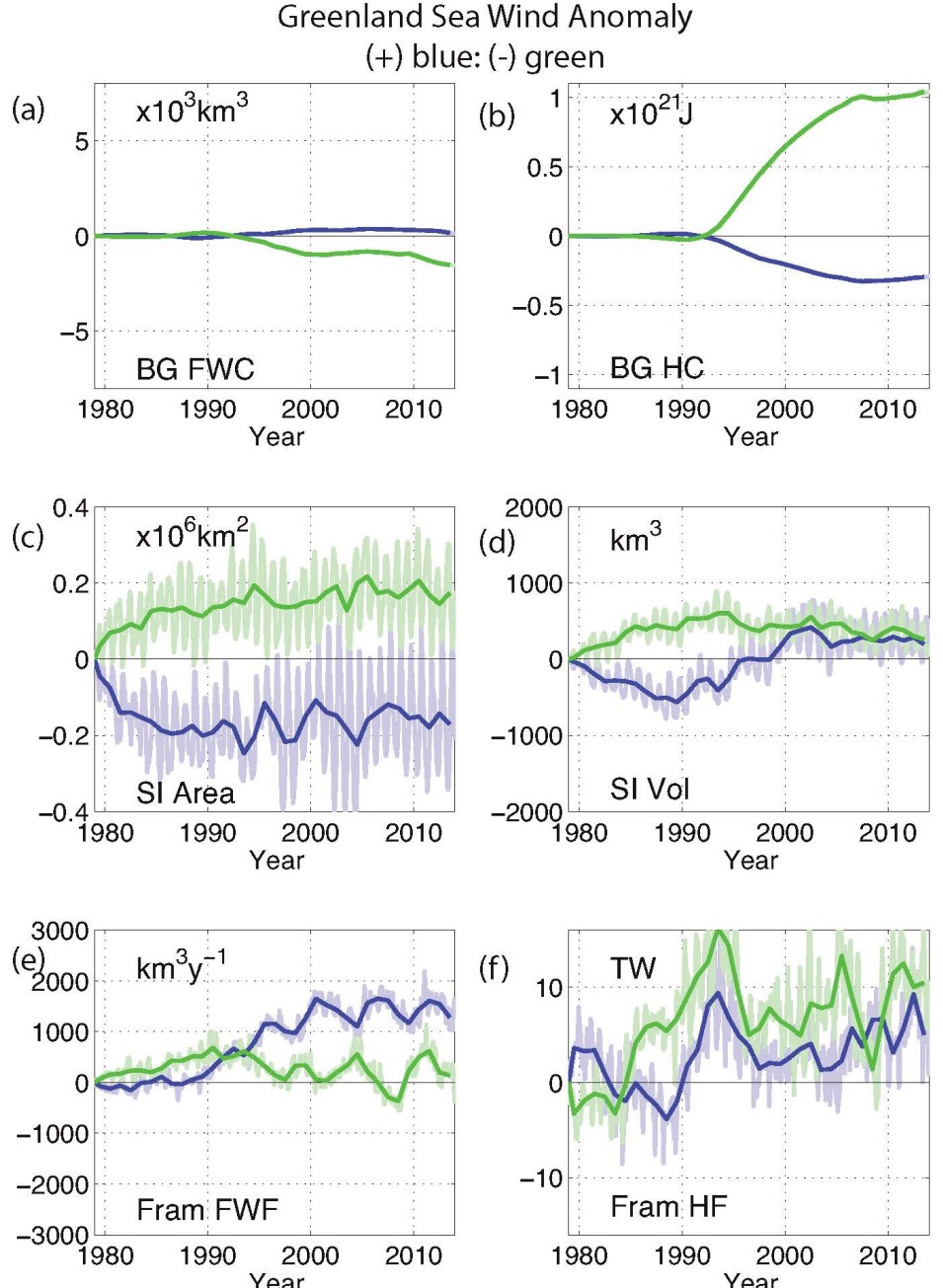

**Figure 7.** CRFs for the Greenland Sea wind anomaly, blue (+) and green (-). Note that the (+) sign implies a stronger cyclonic forcing. The response to a 4HPa surface pressure anomaly (see Fig.5b) is shown of (a) $C_{FWC_{GB}}^{WGS}$, the FWC of the BG (b) $C_{HC_{GB}}^{WGS}$, HC of the BG (c) $C_{SIA}^{WGS}$ , SI area (d) $C_{SIV}^{WGS}$, SI volume (e) $C_{FWF_{Fram}}^{WGS}$, FWF through the Fram Strait and (f) $C_{HF_{Fram}}^{WGS}$, the HF through the Fram Strait.

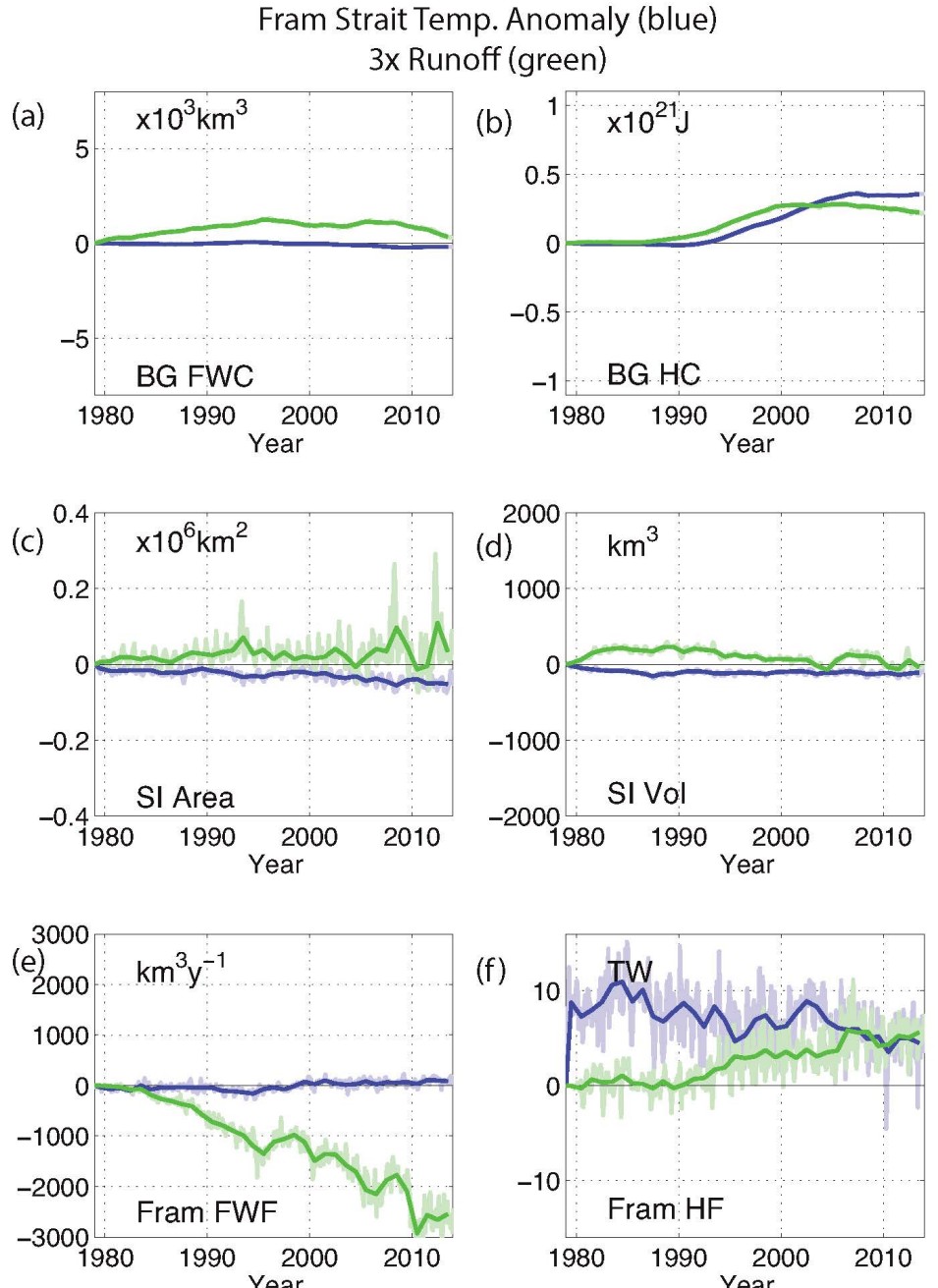

**Figure 8.** CRFs in response to an impulsive 3 x Runoff (green lines) and Fram Strait T (+2C) anomaly (blue lines): (a) the FWC of the BG (b) HC of the BG (c) SI area (d) SI volume (e) FWF through the Fram Strait and (f) the HF through the Fram Strait.

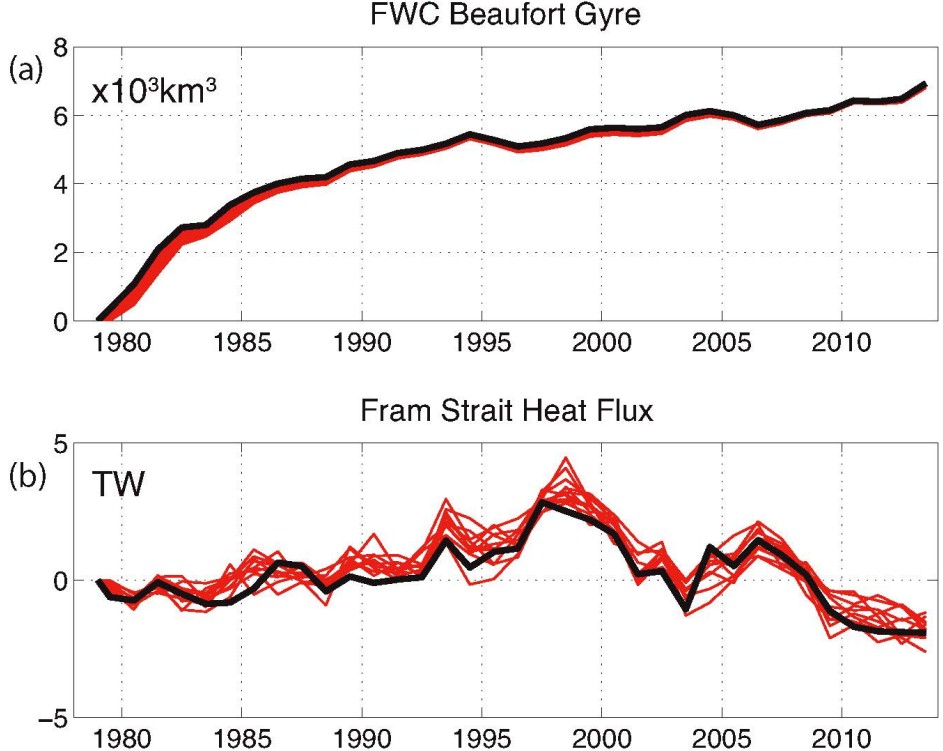

**Figure 9.** CRFs for the BG(+) wind anomaly for (a) the BG FWC and (b) heat flux through the Fram Strait (seasonal cycle removed). Thick black curve is the CRF with the forcing step function anomaly applied on January 1, 1979; ensemble members are show as thin red curves, with the forcing step function applied on February 1, March 1, ..., December 1, 1979.

We adopt the following nomenclature and define $C^{W_{BG}}_{FWC_{BG}}$ (units of m/HPa) here as the response function per unit forcing of the FWC of the BG induced by pressure anomalies (and their associated winds) over the BG, $FWC_{BG}$ (units of m) is the FWC of the BG and $W_{BG}$ (in HPa) is the pressure anomaly over the BG. We may specialise Eq.( 1) to consider the evolution of the FWC of the BG:

$$5 \quad FWC_{BG}(t) = \int_0^t C^{W_{BG}}_{FWC_{BG}}(t - t') \frac{\partial W_{BG}}{\partial t}(t') \, dt' \tag{2}$$

where $W_{BG}$ is the prescribed forcing anomaly (in HPa for the pressure anomaly over the BG).

Imagine now that the BG surface pressure anomaly has oscillatory form thus:

$$W_{BG} = \widehat{W}_{BG} \sin \omega t \tag{3}$$

where $\widehat{W}_{BG}$ is the amplitude of the surface pressure anomaly (in HPa) and $\omega$ is the frequency on which it varies. Let us fit an analytical expression to the FWC BG response function. As can be seen in Fig.6a, it rises on decadal timescales toward a new equilibrium after 30 years or so, but continues to slowly drift upwards. The response to a negative perturbation is (roughly) of opposite sign. The following analytical expression broadly captures the form of $C^{W_{BG}}_{FWC_{BG}}$:

$$5 \quad C^{W_{BG}}_{FWC_{BG}} \times W_{BG_{step}} = \widehat{FWC}_{BG} \left(1 - \exp\left(-\gamma t\right)\right) \tag{4}$$

where the scaling factor $W_{BG_{step}}$ is the magnitude of the step function in the forcing used to construct the CRF and $\widehat{FWC}_{BG}$ is the amplitude of the resulting change in the FWC of the BG. The coefficients $\widehat{FWC}_{BG}$ and $\gamma$ depend on the nature of the forcing and the metric under consideration. They are obtained by fitting the analytical form to the curves shown in the Fig.6a.[7]

The FWC of the BG in response to a forcing can then be written, using Eqs. 2, 3 and 4, and evaluating the integral:

$$10 \quad FWC_{BG}(t) = \frac{\widehat{W}_{BG}}{W_{BG_{step}}} \widehat{FWC}_{BG} \int_0^t \left(1 - \exp -\gamma\left(t - t'\right)\right) \omega \cos \omega t' dt' \tag{5}$$

$$= \frac{\widehat{W}_{BG}}{W_{BG_{step}}} \frac{\widehat{FWC}_{BG}}{\left(1 + \frac{\omega^2}{\gamma^2}\right)} \left(\sin \omega t - \frac{\omega}{\gamma} \left(\cos \omega t - e^{-\gamma t}\right)\right).$$

There are some interesting limit cases:

1. For times much longer than $\gamma^{-1}$, the exponential term dies away and the response oscillates at constant amplitude but shifted in phase relative to the forcing.

2. If $\frac{\omega}{\gamma} << 1$ (low frequency winds) then the response is in phase with the forcing and has an amplitude $\frac{\widehat{W}_{BG}}{W_{BG_{step}}} \widehat{FWC}_{BG}$.

3. If $\frac{\omega}{\gamma} >> 1$ (high frequency winds) then the response is 90 degrees out of phase with the forcing with a much diminished amplitude of $\frac{\gamma}{\omega} \frac{\widehat{W}_{BG}}{W_{BG_{step}}} \widehat{FWC}_{BG}$.

Let us now insert some typical numerical values. Fitting curves to $C^{W_{BG}}_{FWC_{BG}}$, Fig.6a, suggests that $\gamma = \frac{1}{5.7} y^{-1}$ with $\widehat{FWC}_{BG} = 4.9 \times 10^3 km^3$ (the thick green line in Fig.6a corresponding to the negative wind anomaly). We suppose that the
Beaufort High oscillates with an amplitude of $\widehat{W}_{BG} = 6.3$HPa changing in sign with a period of 11 years or so, broadly in accord with observations (see Fig.10b). Then $\omega = \frac{2\pi}{11y} = 0.57 y^{-1}$ and $\frac{\omega}{\gamma} = \frac{0.57}{(1/5.7)} = 3.25 \gtrsim 1$. This suggests that one would expect to see a $90°$ phase lag between the response of the FWC of the BG relative to that of the forcing at these periods with, after the transient has died out, an amplitude of $\frac{\gamma}{\omega} \frac{\widehat{W}_{BG}}{W_{BGstep}} \widehat{FWC}_{BG} = 2.26 \times 10^3 km^3$. The solution, Eq.(5), is plotted in Fig.10a, along with the response function and the wind field so that one can readily ascertain the phase of the response
relative to the forcing. In the first cycle $W_{BG} < 0$ and $FWC_{BG}$ decreases, but lags the forcing by $90°$, or 2.75y if the period of the forcing is 11y. Our analytical prediction (dashed blue line) compares very favourably to that obtained by direct numerical

---

[7] Exponential CRFs are obtained for classical dynamical systems linearised about an equilibrium governed by $\frac{dY}{dt} = -\gamma Y + F(t)$, where $Y$ is the CRF and $F$ is the forcing, yielding a solution $Y = \frac{F}{\gamma}\left(1 - e^{-\gamma t}\right)$. The parameter $\gamma$ can be interpreted as a stability parameter characterising the system which, if linear, is independent of the amplitude of the forcing. See a discussion of $\gamma$ in Manucharyan et al, 2016b.

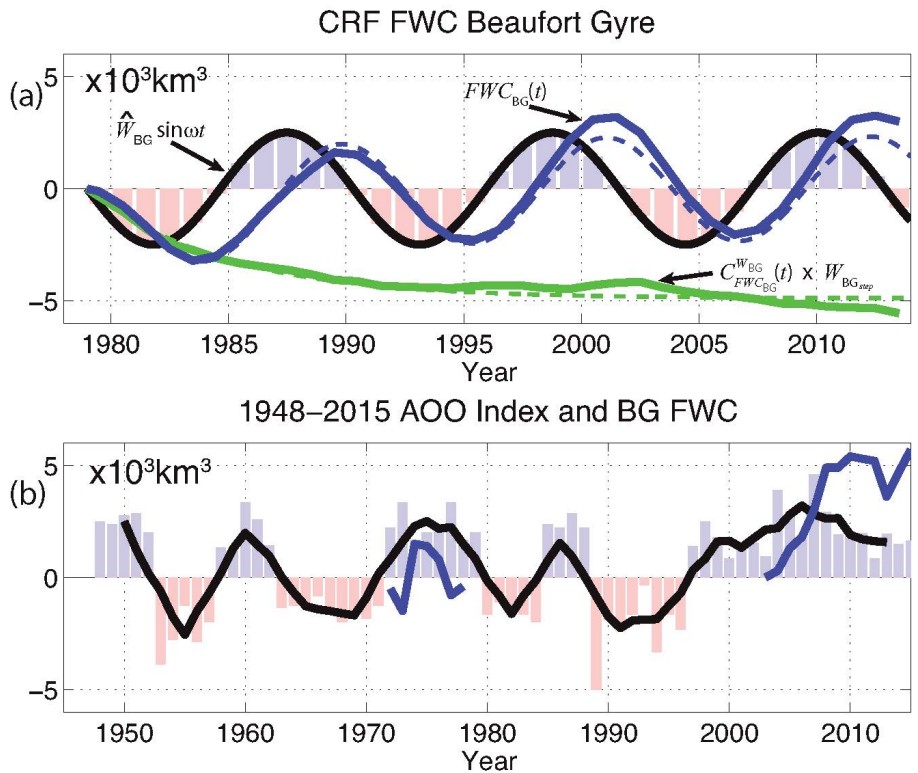

**Figure 10.** (a) Analytical solution (Eq.5) for the response of the FWC of the BG (blue dashed line) to a sinusoidal wind $W_{BG}$ of the form Eq.(3) (thick black line) assuming a response function of the form Eq.(4) (green dashed line, a fit to the thick green line reproduced from Fig.6a). The response of the Arctic GCM to the sinusoidal wind forcing plotted in the same manner in the thick blue line for comparison. (b) The AOO, an index measuring the intensity of the Beaufort High (bars and thick black line), from 1948-2015 (see Proshutinsky et al, 2015). All are 5-year running means. A positive index corresponds to years with an anticyclonic wind stress over the BG and negative are years with a cyclonic wind stress over the BG. The blue line shows observed anomalies of freshwater content (thousands of cubic km) in the BG region.

simulation (thick blue line) in which an oscillating BG wind perturbation was applied to the GCM. This lends strong support to the utility of our approach and the merit of computing CRFs. We now briefly discuss the implications of these results for the observational record of wind forcing and FWC over the BG.

### 3.5.1 Implications for our understanding of decadal variations in the FWC of the Beaufort Gyre

5 The framework we have set up can be used to help us understand how the FWC of the BG has varied over the past few decades. Comparing Figs 6a, 7a & 8a, we see that wind anomalies in the GS region and perturbations to runoff do not significantly affect

$FWC_{BG}$ when compared to changes in the local wind field over the BG. Moreover, if the wind field over the BG oscillates on timescales shorter than the equilibration timescale of the FWC response function, then the FWC will not be in phase with variations in the wind but instead lag it in time.

Fig.10b plots an index of the BG high (the AOO, the Arctic Oscillation Index, a measure of the wind-stress curl integrated over the BG) from 1948 up until 2015 — see Proshutinsky et al. 2015 and legend for more details — which oscillates with a period of 11 years or so, as assumed in the analytical solutions shown in Fig.10a. Also plotted is the FWC from observations from a short period in the 1970s and continued on from 2003. From the early 1990s up until the mid 2000s the anticyclonic driving (as measured by the AOO) over the BG markedly increased. In 2007, the Beaufort high reached a maximum because very strong anticyclonic winds dominated over the gyre throughout the year, decaying in later years. The observed FWC, however, lags the forcing and continues to build, not unlike the prediction obtained from our analytical forcing, plotted in Fig.10a for comparison. One can see that the maximum FWC is observed approximately 3 years after maximum forcing. Of course this is only suggestive — given the short observational record it is impossible to quantitatively check the correctness of the predicted $90°$ lag ($\sim$3 years) between forcing and the BG FWC response to it. Note, for example, that the short observational record in the middle 1970s appears to be in phase with the forcing. That said, it is very unlikely that the FWC can immediately come into equilibrium with the forcing and much more likely to exhibit a lagged response to the wind, as hinted at in the longer observational record starting in 2003 shown Fig.10b.

What is the physics behind the FWC response function setting the timescale $\gamma^{-1}$? At least three important mechanisms come to mind. First the wind-stress curl pumps water down from the surface inflating the freshwater layer. But this occurs in the presence of ice whose ability to communicate the wind stress to the fluid column beneath depends on the nature of the ice pack, a difficult process to model. Perhaps sea ice dynamics is fast relative to $\gamma^{-1}$ whereas slower sea ice thermodynamical processes play more of a role in the CRF timescale. Secondly, baroclinic instability of the BG may be an important mechanism that spreads the FW away laterally, allowing an equilibrium to be achieved (Manucharyan and Spall, 2016a, b). The timescale and equilibrium level at which this is achieved depends on the eddy field which is imperfectly modelled and/or parameterised. Thirdly the availability of fresh water sources and timescales associated with its modification by mixing near continental shelves may come in to play. Thus there is uncertainty in $\gamma$ and $\widehat{FWC_{BG}}$ which controls the detailed response.

## 4   Protocol of proposed perturbation experiments

It would be very interesting to determine how robust the response functions shown in Figs.6, 7 & 8 are across models and understand their dependencies on resolution and physical parameterisation, for example. The CRFs described here are important because, as we have demonstrated, they control and are a summary statement of the response of key Arctic metrics to external forcing. We therefore encourage other groups to carry out such calculations so that we can compare CRFs across many models. Groups would choose their 'best' Arctic simulation (by comparing to observed variables: ice thickness, ice extent, freshwater content, Atlantic water circulation and ability to capture major halocline parameters and Arctic water masses) and perturb it in the manner described in Section 3. The forcings would be identical in all models participating in the CRF experiments. They

are available from the authors, along with recommended protocols for carrying out the experiments, and can be downloaded from the web as described in Section 7. 30-year runs would be required with 5 ensemble members spawned from perturbed initial conditions or by varying the onset timing of the forcing step-function. Monthly-means of $T$, $S$, currents, sea-ice concentration and thickness would be stored and used to compute CRFs. A more detailed account of recommended data output and required model descriptions is also available.

## 5  Conclusions and Expected benefits

Here we have introduced the idea behind and given illustrative examples of Arctic CRFs. They provide a summary statement of how the Arctic responds to the key switches shown in Fig.1. An Arctic model comparison project with CRFs as the organising theme could have many benefits. A focus on the transient response of Arctic models is of direct relevance to Arctic climate change enabling us to engage and overlap with the climate change community. Moreover the framework would enable the project to be informed by, and inform, observations over recent decades and attempts to constrain CRFs by observations would be very profitable. Many different kinds of models could be engaged including ocean-only, coupled, coarse resolution and eddying models, and models with different formulations and physical parameterisations. By doing so the robustness, or otherwise, of CRFs could be determined across a wide range of models and allow different forcing mechanisms to be ranked in order of importance. The 'physics' behind the form of the CRFs would become a natural theme, likely leading to insights into mechanisms underlying Arctic climate change and allowing us to connect to idealised conceptual modelling and theory. In this way the analysis of CRFs can help in the quantitative evaluation of existing hypotheses about Arctic ocean and ice dynamics.

Finally, CRFs could become the building blocks of a physically-based forecast system for the Arctic which harnesses the input of many models to refine the response functions.

## 6  Code availability

The MITgcm is an open source code that can be found online here: http://mitgcm.org/. The 36km Arctic regional model used here is available for public download: http://wwwcvs.mitgcm.org/viewvc/MITgcm/MITgcm_contrib/arctic/cs_36km/

## 7  Data availability

A pdf giving protocol instructions, together with netcdf files containing the forcing fields used in the CRF experiments, can be found here: http://svante.mit.edu/ jscott/FAMOS/Arctic_CRF_Protocol.pdf

*Acknowledgements.* The experiments described here were made possible by support from the NSF program in Arctic Research. J.S. received support from the Joint Program on the Science and Policy of Global Change, which is funded by a number of federal agencies and a consortium of industrial and foundation sponsors. For a complete list please visit http://globalchange.mit.edu/sponsors/all. The comments of

Georgy Manucharyan and an anonymous reviewer are gratefully acknowledged, as are the comments of GMD editor. We would also like to thank the whole FAMOS community who advised and lent their support to this effort.

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
