# Peer review of "'Climate Response Functions' for the Arctic Ocean: a proposed coordinated modelling experiment"

_Geoscientific Model Development, 2016_

## Referee Comment (RC1) · G. Manucharyan (Referee) · 13 Feb 2017

Review of "'Climate Response Functions' for the Arctic Ocean: a proposed coordinated modeling experiment".

The authors propose to conduct a coordinated set of experiments to explore the response of the Arctic Ocean to key external forcing components. The study is motivated by a Green's function approach that allows restoring a linear response of a system to an arbitrary forcing if its response to an impulse forcing is known. The authors provide a comprehensive description of the model experiments that would result in a set of 'Climate Response Functions' (CRFs) for key observables of Arctic circulation and tracer distributions. Using a low-resolution climate model (MITGCM), they successfully demonstrate the usefulness of CRF approach and its potential in improving our understanding of the Arctic Ocean.

I find this study to be very timely, in particular, within a context of an increasing number of hypotheses attempting to explain the freshwater dynamics of the dramatically evolving Arctic Ocean. The manuscript expresses clearly the proposed ideas and methods used and I recommend it for publication after a suggested minor revision aimed at improving its clarity. Below I provide several discussion points, addressing which, would lead to a significant improvement of the manuscript.

1. CRFs can be constructed for any quantitative measure of a model state and as a result, the choice of 'observables' is unlimited. However, it is important to emphasize at least the two key types of 'observables' that can be of use in improving our dynamical understanding: a) 'observables' that are connected to existing hypotheses/theories about the Arctic Ocean dynamics; their CRF's can be directly used to test the existing theories and ii) 'observables' for which CRFs can be constructed from observations and provide a quantitative measure for model skill evaluation. The same logic applies to the choice of forcing.

2. Perhaps a discussion of the expected CRF being the exponential equilibration (Eq.4) will be helpful in the introduction or when Eq. (1) is discussed.

Concerning the discussion on Ln5 p.19: Exponential CRFs are obtained for classical dynamical systems linearized around equilibrium $dY/dt = -\gamma*Y + F(t)$ where $Y$ is observable and $F$ is the forcing. The parameter $\gamma$ can be interpreted as a stability of the observable and for a linear process $\gamma$ should not depend on the amplitude of forcing perturbations. However, the CRF amplitude should be directly proportional to the amplitude of forcing perturbation as well as to $\gamma^{-1}$.

See a discussion of $\gamma$ for the case of Beaufort Gyre freshwater content here: Manucharyan G.E., M.A. Spall, and A.F. Thompson (2016), A theory of the wind-driven Beaufort Gyre variability, J. Phys. Oceanogr., 46, 3263-3278.

2. CRFs are most useful when the system response is linear which is not guaranteed for a finite amplitude forcing perturbations. The manuscript can provide a more detailed discussion about an a priori choice of the amplitude of forcing perturbations; i.e. are there any scaling laws or perhaps observational constraints on when each of the proposed 'observables' responds in a nonlinear way to the 'forcing'?

3. What potential difficulties arise when comparing CRFs from models that have different resolutions and sub-grid-scale parameterizations? What are the key model parameters for a specific observable/forcing pair? In particular, a discussion of the role of mesoscale eddies might be beneficial since the eddy diffusion has been demonstrated to directly affect FWC in idealized Beaufort Gyre models.

4. L 20 p21: the sea ice dynamics responds to wind stress on inertial time scales while its thermodynamics has about 1-2 year time scale. Compared to the 6 year-long FWC equilibration time scale, the sea ice dynamics is sufficiently fast and thus might not directly affect \gamma. I would recommend adding a discussion of the availability of freshwater sources and time scales associated with its modification e.g. due to vertical mixing near continental shelves (note that in a closed Arctic Ocean domain FWC would be preserved unless strong vertical mixing is present).

5. I recommend adding a discussion of an organized data output of post processed CRFs e.g. output frequency, duration of runs etc. In addition to storage of CRF time series, a corresponding list of the key model parameters e.g. ice-ocean-atmosphere drag coefficients, eddy diffusivity scheme, vertical tracer mixing, momentum dissipation/bottom drag, etc.. would ease the subsequent analysis of results.

6. Finally, I recommend emphasizing that the analysis of various CRFs can help in the quantitative evaluation of existing hypotheses about the Arctic Ocean dynamics.

---

## Referee Comment (RC2) · Anonymous Referee #2 · 4 Mar 2017

This paper proposes a coordinated set of Arctic modelling experiments to look at how the Arctic might respond to various forms of external climate forcing. The different forms of climate forcing considered is wind anomalies (over the Beaufort as well as the Greenland seas), runoff and gateway inflow (Bering, Fram Strait). The authors explore an approach to get at the linear response to step changes in forcing through a convolution approach – the climate response functions in the title. The authors include some preliminary analysis of the idea using experiments with the MIT model.

This is an interesting topic and the community can use more well planned and coordinated experiments. Understanding Arctic climate variability is important and thus the approach suggested in this study is worth considering. Therefore the manuscript is an

appropriate subject for publication in GMD. It is generally well written and easy to follow. That said, some small changes could be made to improve the manuscript before publication.

To start with, a very minor point, but given this is a European journal, I'm surprised by the American spelling of modelling. Would much prefer to see the proper English spelling with 2 l's in the title, and through the text.

Figure 1 caption: The background colour shading is bathymetry (and elevation over land), but this is not mentioned in the caption.

Pressure units: Doesn't GMD request the use of metric units? If so, please change mbar to HPa everywhere.

Page 4, lines 26-28: The sentence, with the multiple dashes, is a bit too broken up to easily follow. Given the importance of the linearity of the response, this point can be expanded upon.

Section 2.2.1 – It lists the key switches. But it would be good to add a bit more motivation on why they were chosen.

Page 5, line 14 – Please add a reference related to the lack of near surface observations.

Section 2.2.2 – Would like to see a bit more detail in the discussion of metrics. Strength of boundary currents – which ones, where should they be measured, etc. For the ice fields, what domain should they be averaged over? Same for the mixed layer. And isn't the flux through various straits the same as the export of heat and freshwater, since that export only occurs through the gateway straits. Some of this can be answered by linking the text better figure 2a, which may have the necessary answers in a graphical form.

Figure 2b – Why the given box? Doesn't seem tightly tied to the inflow or the warmest temperatures. Figure 3 caption – define the negative sign for the fluxes.

Page 10, line 3 – Ilicak, not Iliac for the reference.

Section 3.2.4 – Given that temperature and salinity vary by section and season, won't fixed T and S changes always lead to some density compensation?

Section 3.3 – Are the CRFs applied all together, or individually? I think the latter, but the text isn't 100% clear on this.

Page 13, line 11 – I don't necessarily see a new equilibrium in the figure. But is an equilibrium necessary?

Section 3.3, summary point 3 – The CRFs are symmetric with respect to some metrics, but not all. Maybe make that clear in the text here.

Section 3.4 – Why does the heat flux through Fram Strait have a much larger envelope?

Page 14, line 23 – Do you mean not many ensembles, or not many members?

Figure 8 – The two changes shown here are not really opposite, so showing them together is not consistent with the previous figures. Even though it adds a figure, it might be good to separate the results from these two switches into two separate figures.

Page 21, line 22 – But might freshwater processes from river runoff depend on resolution and the processes involved in shelf basin exchange? Such a question may prompt the idea that it might be good to have these CRF experiments done with different resolutions as well as different models to get at this question. Maybe add some discussion of that point?

Page 22, line 1 – Give the FAMOS web link here too.

Section 5 – Not sure I like a conclusion that is just a numbered listed. Some more explanation, especially of the hoped significance of this coordinated experiment, would be good.

---

## Author Comment (AC1) · 6 Mar 2017

We thanks Georgy Manucharyan for his encouraging comments and will address them all in revision.

Motivation of CRFs

We will indeed motivate our choice of CRFs more comprehensively and suggestions in 1a and 1b on how to do that are well taken. We will also make much more use of Georgy's analytical model of equilibration of the BG to motivate Eq.4 of our manuscript. This enables us to clearly connect through to mechanisms.

Linearity

Linearity of our CRFs is critical if we are to use convolution theory in a predictive way. If we drive our system with very large wind anomalies then the response is not linear and less symmetric wrt change in sign of the forcing anomaly. However, for 'realistic' amplitudes of wind forcing the response is indeed 'usefully' linear, as we attempt to demonstrate in the paper. We will make this much clearer in revision.

Model resolution and parameterization

We believe that the models will be sensitive to resolution and parameterization and this is one of the motivations for our study. For example, resolution may be important in equilibrating the FWC of the BG either due to the ability to resolve eddies, but also boundary current and 'outward' (as well as inward) pathways.

Availability of FW to BG

This issue is an important one and indeed demands a more detailed discussion. This will be added in revision.

Data output and postprocessing

A fuller description will be given in the body of the paper but we prefer to devolve this to the associated 'protocol' writeup that goes along with the paper. This is still being fleshed out after discussion with the FAMOS group. We will look in to also posting this on the GMD website.

Testing hypotheses and mechanisms

we agree that CRFs are very valuable in this regard, a point which will be expanded upon in revision.

---

## Author Comment (AC2) · 6 Mar 2017

We thanks the reviewer for their helpful comments.

English spelling and metric units

I have lived in the US for too long and will revert back to English spelling :-). We will also connect to metric units in revision.

Captions to Fig.1

Thanks for pointing this out - we will improve the caption in revision.

Linearity

[Figure]

This was picked up also in the Manucharyan review. Needless to say, not all our CRFs are linear and the degree of linearity depends on the magnitude of the forcing and the particular CRF being plotted. FWC in the BG, for example, appears to be linear in our model for moderate forcing amplitudes, but is not when the wind anomalies are too large. Much more will be written on this issue in revision.

Some metrics are indeed more linear (and symmetric wrt sign of the forcing) - this will be commented on in revision although we do not yet understand all the issues here.

Key Switches

We will motivate our choice of key switches in more detail in revision. The wind patterns correspond, roughly, to the leading modes of atmospheric variability driving the Arctic - the AO (AOO) and the Icelandic/Greenland low. Heat transport through Fram strait and freshwater anomalies are also key drivers of Arctic climate variability and change.

Discussion of metrics and lack of near-surface observations

We are discussion with Andrey Proshutinsky and these points will be addressed in revision.

Designation of anomalies of Fram Strait transport

We chose the box in Fig.2b to roughly encompass the major temperature signal in the section. We can revisit our detailed choice. We do not completely understand the comment on the seasonal cycle. Suffice to say our (salinity-compensated) T anomaly does not have a seasonal cycle and is meant to represent a secular trend in the properties of Atlantic water entering the Arctic.

Application of CRFs

The CRFs are applied individually, but they could be added sequentially and/or simultaneously. Depending on the linearity, we could de-convolve the separate effects. This will be clarified in revision. As you surmised, the number of ensemble members is

being refereed to, not the number of ensembles. In our coarse resolution experiments with the MITgcm, we found that small numbers of ensemble members is required because most of the variability is forced, rather than internal to the model. This may not be true of other models, particularly when such models are run at higher resolution and thus presumably exhibit greater internal variability.

Figure 8.

Yes, it would indeed be best to separate them. We confuse not only the reader but ourselves!

Model resolution and paramterization

Model differences will surely impact our CRFs, but this is precisely a key reason we are interested in comparing them and studying them. For example we are already finding (not reported in the paper) that CRFs, although having broadly similar characteristics, differ in detail across models, both in amplitude and timescale of response. Our goal is to understand why and how, and what they might look like in the real system.

Conclusion

We agree that the conclusion could be much better done - and is an important job in revision.

---

## Short Comment (SC1) · 10 Mar 2017

Georgy's comment #1 (and #6) reverberates in particular with me, and we will definitely make this point. There is clearly a need for models with high levels of skill in the community, but at the end of the day we also need to do good hypothesis-testing based science. These objectives are not completely independent but we need to have a clear strategy on observables for both.

Georgy has helpful comments and insights about the BG FWC results, and adding additional science discussion here will strengthen the paper, using this question as an example of hypotheses to explore. However I might suggest we probably cannot go into as much specific detail as the reviewer might desire, but this might make for an

interesting side paper. Hopefully this intercomparison project will spawn exploration of many additional science-based ideas and published results.

John mentioned linearity; to be more explicit, we indeed tested a larger wind anomaly, but it strongly affected the large-scale circulation in the Arctic in an unrealistic manner. I think our choice is reasonably well motivated by observations (we will make this clearer) and results appear mostly linear for many observables. While we plan to tackle what defines "mostly", it is beyond the scope and purpose of this submission.

Finally, yes, for point #5 we plan to augment our official CRF 'protocol' with details on model output, and agree that we need to maintain a database of specific model details and parameters.

———————————————————

---

## Short Comment (SC2) · 10 Mar 2017

We thank the reviewer for these helpful comments and suggestions.

The reviewer makes a good point that the presentation of fig 2 and the motivation for the switches and gateway straits is not as well presented and motivated as it could be. As John noted, we are interested in examining how different models respond to climate change, and to understand the underlying processes and mechanisms. Thus we are all on the same page that the flux through various straits is important to diagnose. Measuring and comparing boundary currents themselves is a bit more problematic across a spread of different models (grids and resolution), but that being said we are certainly open to additional important metrics we may have missed and/or thoughts

about how best to both assess Arctic change and compare across models.

To answer the reviewer's question about ensemble HF through Fram St: FWC is an integrated quantity in x,y,z whereas the HF is computed through a section, the latter quite sensitive to interannual forcing among other model setup choices, so it is not surprising on general grounds to see a larger envelope. We agree that the topic of ensembles is important, which is why we include section 3.4, but a more complete exploration of ensemble results is beyond the scope of this specific paper. But we will add more discussion on the Fig 9 results.

————————————————————

---

## Short Comment (SC3) · 11 Mar 2017

4. Availability of freshwater sources and mixing: I think that this is one of important problems for BG simulations and especially in idealized experiments in the closed basins. In the real world it is very possible that saturation of the BG freshwater content could not be reached because of freshwater permanently coming from rivers, precipitation and melting ice and due to mixing. We have a paper in preparation for JGR-Oceans with more detailed discussion about these processes based on both modeling and observations.

6. I think that we have already mentioned this in the paper but I agree that we have to emphasize this for clarity.

---

## Author Response (AR1)

Review of "'Climate Response Functions' for the Arctic Ocean: a proposed coordinated modeling experiment".
Referee 1 (G. Manucharyan)

The authors propose to conduct a coordinated set of experiments to explore the response of the Arctic Ocean to key external forcing components. The study is motivated by a Green's function approach that allows restoring a linear response of a system to an arbitrary forcing if its response to an impulse forcing is known. The authors provide a comprehensive description of the model experiments that would result in a set of 'Climate Response Functions' (CRFs) for key observables of Arctic circulation and tracer distributions. Using a low-resolution climate model (MITGCM), they successfully demonstrate the usefulness of CRF approach and its potential in improving our understanding of the Arctic Ocean.

I find this study to be very timely, in particular, within a context of an increasing number of hypotheses attempting to explain the freshwater dynamics of the dramatically evolving Arctic Ocean. The manuscript expresses clearly the proposed ideas and methods used and I recommend it for publication after a suggested minor revision aimed at improving its clarity. Below I provide several discussion points, addressing which, would lead to a significant improvement of the manuscript.

1.CRFs can be constructed for any quantitative measure of a model state and as a result, the choice of 'observables' is unlimited. However, it is important to emphasize at least the two key types of 'observables' that can be of use in improving our dynamical understanding: a) 'observables' that are connected to existing hypotheses/theories about the Arctic Ocean dynamics; their CRF's can be directly used to test the existing theories and ii) 'observables' for which CRFs can be constructed from observations and provide a quantitative measure for model skill evaluation. The same logic applies to the choice of forcing.

We have motivated our choice of CRFs more comprehensively and included the suggestions above. See revisions in Section 2.2.2.

2a. Perhaps a discussion of the expected CRF being the exponential equilibration (Eq.4) will be helpful in the introduction or when Eq. (1) is discussed.

Concerning the discussion on Ln5 p.19: Exponential CRFs are obtained for classical dynamical systems linearized around equilibrium $dY/dt = -\gamma*Y + F(t)$ where $Y$ is observable and $F$ is the forcing. The parameter $\gamma$ can be interpreted as a stability of the observable and for a linear process $\gamma$ should not depend on the amplitude of forcing perturbations. However, the CRF amplitude should be directly proportional to the amplitude of forcing perturbation as well as to $\gamma^{-1}$.

See a discussion of $\gamma$ for the case of Beaufort Gyre freshwater content here: Manucharyan G.E., M.A. Spall, and A.F. Thompson (2016), A theory of the wind-driven Beaufort Gyre variability, J. Phys. Oceanogr., 46, 3263-3278.

We have included a footnote (on page 20) inspired by this suggestion, and included a reference to the above paper.

2b. CRFs are most useful when the system response is linear which is not guaranteed for a finite amplitude forcing perturbations. The manuscript can provide a more detailed discussion about an a priori choice of the amplitude of forcing perturbations; i.e. are there any scaling laws or perhaps

observational constraints on when each of the proposed 'observables' responds in a nonlinear way to the 'forcing'?

Linearity of our CRFs is critical if we are to use convolution theory in a predictive way. If we drive our system with very large wind anomalies then the response is not linear and less symmetric wrt change in sign of the forcing anomaly. However, for 'realistic' amplitudes of wind forcing the response is indeed 'usefully' linear, as we attempt to demonstrate in the paper. We will make this much clearer in revision.

We indeed tested a larger wind anomaly (20HPa), but it strongly affected the large-scale circulation in the Arctic in an unrealistic manner. Our choice of 6HPa is motivated by observations and results appear linear for many observables, but not all. While we plan to tackle what defines "mostly", it is beyond the scope and purpose of this paper.

3. What potential difficulties arise when comparing CRFs from models that have different resolutions and sub-grid-scale parameterizations? What are the key model parameters for a specific observable/forcing pair? In particular, a discussion of the role of mesoscale eddies might be beneficial since the eddy diffusion has been demonstrated to directly affect FWC in idealized Beaufort Gyre models.

We believe that the models will be sensitive to resolution and parameterization and this is one of the motivations for our study. For example, resolution may be important in equilibrating the FWC of the BG either due to the ability to resolve eddies, but also boundary current and 'outward' (as well as inward) pathways. Comparing models of different resolutions presents no specific difficulty.

4. L 20 p21: the sea ice dynamics responds to wind stress on inertial time scales while its thermodynamics has about 1-2 year time scale. Compared to the 6 year-long FWC equilibration time scale, the sea ice dynamics is sufficiently fast and thus might not directly affect \gamma. I would recommend adding a discussion of the availability of freshwater sources and time scales associated with its modification e.g. due to vertical mixing near continental shelves (note that in a closed Arctic Ocean domain FWC would be preserved unless strong vertical mixing is present).

I think that this is one of important problems for BG simulations and especially in idealized experiments in the closed basins. In the real world it is very possible that saturation of the BG freshwater content could not be reached because of freshwater permanently coming from rivers, precipitation and melting ice and due to mixing. See text inserted around line 30, pg 21.

5. I recommend adding a discussion of an organized data output of post processed CRFs e.g. output frequency, duration of runs etc. In addition to storage of CRF time series, a corresponding list of the key model parameters e.g. ice-ocean-atmosphere drag coefficients, eddy diffusivity scheme, vertical tracer mixing, momentum dissipation/bottom drag, etc.. would ease the subsequent analysis of results.

We prefer to devolve very detailed (but nevertheless important) 'protocol' discussions to the protocol write-up that goes along with the paper. This has been posted to the FAMOS website and will be posted to the GMD website if thought appropriate.

We have augmented our official CRF 'protocol' with details on model output, and begun a database of specific model details and parameters.

6. Finally, I recommend emphasizing that the analysis of various CRFs can help in the quantitative evaluation of existing hypotheses about the Arctic Ocean dynamics.

We agree that CRFs are very valuable in this regard, a point that has been emphasized in revision – see Conclusions.

Referee 2 (anonymous)

This paper proposes a coordinated set of Arctic modelling experiments to look at how the Arctic might respond to various forms of external climate forcing. The different forms of climate forcing considered is wind anomalies (over the Beaufort as well as the Greenland seas), runoff and gateway inflow (Bering, Fram Strait). The authors explore an approach to get at the linear response to step changes in forcing through a convolution approach – the climate response functions in the title. The authors include some preliminary analysis of the idea using experiments with the MIT model.

This is an interesting topic and the community can use more well planned and coordinated experiments. Understanding Arctic climate variability is important and thus the approach suggested in this study is worth considering. Therefore the manuscript is an appropriate subject for publication in GMD. It is generally well written and easy to follow.

That said, some small changes could be made to improve the manuscript before publication.

To start with, a very minor point, but given this is a European journal, I'm surprised by the American spelling of modelling. Would much prefer to see the proper English spelling with 2 l's in the title, and through the text.

Modelling has now been correctly spelt and we have also adopted metric units in revision, HPa instead of mb.

Figure 1 caption: The background colour shading is bathymetry (and elevation over land), but this is not mentioned in the caption.

Caption improved.

Pressure units: Doesn't GMD request the use of metric units? If so, please change mbar to HPa everywhere.

Done.

Page 4, lines 26-28: The sentence, with the multiple dashes, is a bit too broken up to easily follow. Given the importance of the linearity of the response, this point can be expanded upon.

Sentence has been rewritten.

Linearity was picked up also in the Manucharyan review. Needless to say, not all our CRFs are linear and the degree of linearity depends on the magnitude of the forcing and the particular CRF being plotted. FWC in the BG, for example, appears to be linear in our model for moderate forcing amplitudes, but is not when the wind anomalies are too large. Some metrics are indeed more linear (and symmetric wrt sign of the forcing) - this is commented on in revision although we do not yet understand all the issues..

Section 2.2.1 – It lists the key switches. But it would be good to add a bit more motivation on why they were chosen.

The reviewer makes a good point that the presentation of fig 2 and the motivation for the switches and gateway straits is not as well presented and motivated as it could be. We are interested in examining how different models respond to climate change, and to understand the underlying processes and mechanisms. Thus we are all on the same page that the flux through various straits is important to diagnose.

We will motivate our choice of key switches in much more detail in revision. The wind patterns correspond, roughly, to the leading modes of atmospheric variability driving the Arctic - the AO (AOO) and the Icelandic/Greenland low. Heat transport through Fram strait and freshwater anomalies are also key drivers of Arctic climate variability and change.

Page 5, line 14 – Please add a reference related to the lack of near surface observations.

Done – see line 23, page 6.

Section 2.2.2 – Would like to see a bit more detail in the discussion of metrics. Strength of boundary currents – which ones, where should they be measured, etc. For the ice fields, what domain should they be averaged over? Same for the mixed layer. And isn't the flux through various straits the same as the export of heat and freshwater, since that export only occurs through the gateway straits. Some of this can be answered by linking the text better figure 2a, which may have the necessary answers in a graphical form.

Measuring and comparing boundary currents across a spread of different models (with variable grids and resolution) should be carried out, but we have focused here on fluxes through Straits. That said we are certainly open to inclusion of additional important metrics we may have missed and/or thoughts about how best to both assess Arctic change and compare across models.

In revision we have tried to make more use of Fig.2a in our discussion.

Figure 2b – Why the given box? Doesn't seem tightly tied to the inflow or the warmest temperatures.

We chose the box in Fig.2b to roughly encompass the major temperature signal in the section. We can revisit our detailed choice. We do not completely understand the comment on the seasonal cycle. Suffice to say our (salinity-compensated) T anomaly does not have a seasonal cycle and is meant to represent a secular trend in the properties of Atlantic water entering the Arctic.

Figure 3 caption – define the negative sign for the fluxes.

Done

Page 10, line 3 – Ilicak, not Iliac for the reference.

Done

Section 3.2.4 – Given that temperature and salinity vary by section and season, won't fixed T and S changes always lead to some density compensation?

We have applied the S compensation on monthly-means and this seems to readily maintain density compensation.

Section 3.3 – Are the CRFs applied all together, or individually? I think the latter, but the text isn't 100% clear on this.

The CRFs are applied individually, but they could be added sequentially and/or simultaneously. Depending on the linearity, we could de-convolve the separate effects. This will be clarified in revision. As you surmised, the number of ensemble members is being referred to, not the number of ensembles. In our coarse resolution experiments with the MITgcm, we found that small numbers of ensemble members is required because most of the variability is forced, rather than internal to the model. This may not be true of other models, particularly when such models are run at higher resolution and thus presumably exhibit greater internal variability

Page 13, line 11 – I don't necessarily see a new equilibrium in the figure. But is an equilibrium necessary?

No, an equilibrium after 30 years is not required or perhaps even expected.. The rising trend may be particular to this model. Point noted in 1$^{st}$ paragraph of Section 3.3.

Section 3.3, summary point 3 – The CRFs are symmetric with respect to some metrics, but not all. Maybe make that clear in the text here.

See point 3, page 15.

Section 3.4 – Why does the heat flux through Fram Strait have a much larger envelope?

FWC is an integrated quantity in x,y,z whereas the HF is computed through a section, the latter quite sensitive to interannual forcing among other model setup choices, so it is not surprising on general grounds to see a larger envelope. We agree that the topic of ensembles is important, which is why we include section 3.4, but a more complete exploration of ensemble results is beyond the scope of this specific paper.

Page 14, line 23 – Do you mean not many ensembles, or not many members?

Corrected, thanks.

Figure 8 – The two changes shown here are not really opposite, so showing them together is not consistent with the previous figures. Even though it adds a figure, it might be good to separate the results from these two switches into two separate figures.

We have thought about this and decided to keep the figure as is.

Page 21, line 22 – But might freshwater processes from river runoff depend on resolution and the processes involved in shelf basin exchange? Such a question may prompt the idea that it might be good to have these CRF experiments done with different resolutions as well as different models to get at this question. Maybe add some discussion of that point?

Model differences and model resolution will surely impact our CRFs, but this is precisely a key reason we are interested in comparing them and studying them. For example we are already finding (not reported in the paper) that CRFs, although having broadly similar characteristics, differ in detail across models,

both in amplitude and timescale of response. Our goal is to understand why and how, and what they might look like in the real system.

Page 22, line 1 – Give the FAMOS web link here too.

Link added.

Section 5 – Not sure I like a conclusion that is just a numbered listed. Some more explanation, especially of the hoped significance of this coordinated experiment, would be good.

We agree that the conclusion could have been much better done: rewritten.

[revised manuscript text omitted]

---

## Author Response (AR2)

Dear Sophie,

Thanks for your careful reading of our manuscript. We have uploaded a new pdf and diff (relative to the last version) to help you view what we have done.

Below I briefly respond to your comments.

During revision we have also made some further minor adjustments to the text to increase readability.

Many thanks for your consideration.

John Marshall

Topical Editor Decision: Publish subject to minor revisions (Editor review) (22 May 2017) by Sophie Valcke
Comments to the Author:
Dear Authors,

Thanks you very much for your updated manuscript that answers the referee's comments. But before publishing the manuscript, I would ask you to consider the following additional comments:

-Regarding the density compensation discussed on p.12, it is not clear to me how you got to S+0.253psu. In the footnote, can you also give the equation, not only the parameter values? Also in your reply to referee 2, you wrote "this seems to readily maintain density compensation"; can you be more precise how you got to that conclusion and give some details in the paper?

We have corrected and expanded footnote 6 in the paper.

-On p. 19, L1, please change "units of HPa" for units of "m/HPa" (I think)

done

-In Figure 10 captions, please described the thick green line (and maybe add "is" before "plotted"?)

done

-In section 4, please refer to section 7, when you mention "from the web" and/or when you discuss the recommended data output.

done

-In section 6, you need to specify the version of the code used here.

done

Minor comments

- The first sentence of section 2.2.2 needs to be reformulated; without the text between hyphens and in parentheses, it gives: "Ideal observables are integrated quantities should be constrained by observations, indicative of underlying mechanisms and of climatic relevance" which does not look right to me. Please rephrase this sentence.

done

-In the captions of Figure 6 & 7, please replace "4mb" by "4Hpa"

done

-On figures 6, 7 and 8, please use either the abbreviations everywhere or the full expressions. For example, the Fresh Water Content appears as FWC but the heat content is spelled out. It could be useful to consistently give the full expressions in the captions and only the abbreviations on the figures per se.

done

-on p.22 L 23, please change "depends of the eddy field" for "depends on the eddy field"

done

[revised manuscript text omitted]